# Hydrogenolysis of Glycerol on the ZrO$_2$-TiO$_2$ Supported Pt-WO$_x$ Catalyst

**Zhiwen Xi [1], Zhe Hong [1], Fangtao Huang [1], Zhirong Zhu [1,*] , Wenzhi Jia [2,*] and Junhui Li [3]**

[1] Shanghai Key Lab of Chemical Assessment and Sustainability, School of Chemical Science and Engineering, Tongji University, Shanghai 200092, China; xizhiwen007@sina.com (Z.X.); hongzhe0816@163.com (Z.H.); FangtaoHuang1993@163.com (F.H.)

[2] Department of Materials Engineering, Huzhou University, Huzhou 313000, China

[3] School of Chemical Engineering, Xiangtan University, Xiangtan 411105, China; canjian.12@163.com

\* Correspondence: zhuzhirong@tongji.edu.cn (Z.Z.); jiawenzhi@126.com (W.J.)

**Abstract:** A series of Pt/WOx-ZrO$_2$-TiO$_2$ catalysts with different Ti/Zr molar ratios was prepared by an evaporation induced self-assembly method, and used to efficient hydrogenolysis of glycerol to 1-PO and 1,3-PDO. BET, XRD, Raman, TEM, XPS and Py-IR were employed to characterize the physicochemical properties of the catalysts. The structural and acidic properties of the catalysts were affected by the Ti/Zr ratio of the support ZrO$_2$-TiO$_2$. Two new crystalline phases of ZrTiO$_4$ and Ti$_2$ZrO$_6$ and the amount of acid sites were detected in the Pt/WOx-ZrO$_2$-TiO$_2$ catalysts. 1-PO is dominant in all products of glycerol hydrogenolysis over the supported Pt-WOx catalysts, which is attributed to more Lewis acid sites on the catalyst surface. The Pt/WOx-ZrO$_2$-TiO$_2$ catalyst with a Ti/Zr ratio of 7/3 showed the highest 1,3-PDO yield (25.3%) and 1-PO yield (42.3%), due to its more acid sites including Brønsted and Lewis, and higher concentration of surface Pt$^0$.

**Keywords:** hydrogenolysis of glycerol; 1,3-propanediol; 1-propanediol; Pt-WO$_x$; ZrO$_2$-TiO$_2$

## 1. Introduction

With the fossil resources depleting and concerns about environmental issues increasing, the utilization of biomass as a sustainable alternative rapidly pays people's attention. Glycerol is the main by-product from biodiesel plants. The transformation of glycerol to value-added chemicals through reforming, dehydration, oxidation, esterification or hydrogenolysis has gained increasing investigations from academic and industrial researchers. Hydrogenolysis of glycerol to 1-propanediol (1-PO) and 1,3-propanediol (1,3-PDO) is considered as a promising route for the utilization of glycerol. 1-Propanol (1-PO) is utilized mainly as an industrial solvent, printing ink and chemical intermediate for the synthesis of *n*-propyl acetate [1]. 1,3-PDO is one of the most value-added derivatives from glycerol in that it is an important chemical for the manufacture of polytriethylene terephthalate, polyurethanes and cyclic compounds with wide applications [1].

Numerous works have been done towards glycerol hydrogenolysis to 1,2-PDO and 1,3-PDO in recent years. Up to date, main kinds of catalysts used for the hydrogenolysis of glycerol include noble metal based, Cu-based and Ni-based catalysts, one of which the noble metal based catalysts are concerned hotly, such as Rh-ReO$_x$/SiO$_2$ [2], Ru-Re/SiO$_2$ [3–5], Ir-ReO$_x$/SiO$_2$ [6,7], Pt-H$_4$SiW$_{12}$O$_{40}$/ZrO$_2$ [8,9] and Pt-WO$_3$/ZrO$_2$ [10,11]. Among these catalysts, the Pt-WO$_3$ system catalysts were mostly investigated due to their potential application prospect. For the Pt-WO$_3$ system catalysts, 1,3-PDO was the focus of the product of glycerol hydrogenolysis, and a bi-functional mechanism comprising the synergism of metal sites and acid sites has been universally proposed by many researchers for the reaction path of glycerol hydrogenolysis to 1,3-PDO [8–12]. However, the

researches corresponding to the transformation of glycerol hydrogenolysis toward two valuable 1-PO and 1,3-PDO are relatively less among the present literatures.

Undoubtedly, the acidity of the catalysts is fundamental in the glycerol hydrogenolysis [13]. To change the activity and selectivity in glycerol hydrogenolysis, the many routes to modify the acidic properties of catalysts always are the hot spot. Besides, the effect of Brønsted and Lewis acid sites on the reactivity of glycerol hydrogenolysis was continually reported. It was summarized by Fan [14] that the role of Brønsted acid sites (M–OH) was to protonate the secondary hydroxyl group of glycerol directly and then eliminate this group, or to activate glycerol via the etherification reaction to form the M–O–$CH_2CH(OH)CH_2OH$ surface species, which favored the depletion of the secondary hydroxyl group of glycerol. It was considered that the acidic properties of $WO_3$-$ZrO_2$ were dependent on the condensation states of the $WO_x$ species (domain size), which were affected by several parameters like the W contents, calcination temperature and preparation method [15,16]. Strong Brønsted acidity was obtained when the $WO_x$ species were well dispersed and reached monolayer coverage on the $ZrO_2$ surface [17].

In relation to supports, acidic supports with diverse textural properties such as $SiO_2$-$Al_2O_3$, zeolites, $Al_2O_3$, $ZrO_2$, $TiO_2$, $Nb_2O_5$, activated carbons, etc. have been used to prepare noble metal catalysts for the hydrogenolysis of glycerol because of the critical functions of the acidic and textural properties of the supports on this reaction [18–21]. Binary oxides like $ZrO_2$-$Al_2O_3$ [22], $TiO_2$-$Al_2O_3$ [23] and $ZrO_2$-$TiO_2$ [24] have been used widely due to their unique acid-base properties, high specific surface area along with good thermal stability. In fact, an improved acidic property was expected when the $WO_x$ species were loaded on these composite oxides due to the modified interaction between active species and the support. Additionally, as a result, the acidic catalytic performance could be enhanced compared to the single $ZrO_2$ or $TiO_2$ supported $WO_x$ catalysts.

Herein, a series of $ZrO_2$-$TiO_2$ composite oxides with different Ti/Zr ratios as the supports of Pt-WOx based catalysts were prepared via an Evaporation Induced Self Assembly (EISA) method in one step. BET, XRD, Raman, XPS, Py-IR and TEM were employed to characterize the physicochemical properties of the Pt/WOx-$ZrO_2$-$TiO_2$ catalysts. Compared with $ZrO_2$ and/or $TiO_2$ supported Pt-WOx catalysts, the Pt/WOx-$ZrO_2$-$TiO_2$ catalysts have higher surface area, higher concentration of $Pt^0$ and more Brønsted and Lewis acid sites. It is interestingly found that the Pt/WOx-$ZrO_2$-$TiO_2$ catalyst with a Ti/Zr ratio of 7/3 (PtWZr3Ti7) catalyst has the highest activity, with a 1,3-PDO yield of 25.3% and a 1-PO yield of 42.3%.

## 2. Results and Discussions

### 2.1. Characterization

The $N_2$ physisorption isotherms and pore size distributions of the Pt/WOx-$ZrO_2$-$TiO_2$ (PtWZrTi) catalysts with different Ti/Zr molar ratio are shown in Figure 1. As can be seen from Figure 1A, the samples show a type IV adsorption isotherm but had a different hysteresis loop. The Pt-$WO_x$/$ZrO_2$ (PtWZr) catalyst exhibited an H4 hysteresis loop, while H2 hysteresis loop was presented in the Pt-$WO_x$/$TiO_2$(PtWTi) and PtWZrTi catalyst. From Figure 1B, the average pore size distribution was around 2-5 nm for all the samples. The physicochemical properties including surface area, pore size and pore volume, are listed in Table 1. The Pt/WOx-$ZrO_2$-$TiO_2$ catalyst with a Ti/Zr molar ratio of 7/3 (PtWZr3Ti7) possessed a high surface area (64.4 $m^2$ $g^{-1}$) and pore volume (0.083 $cm^3$ $g^{-1}$).

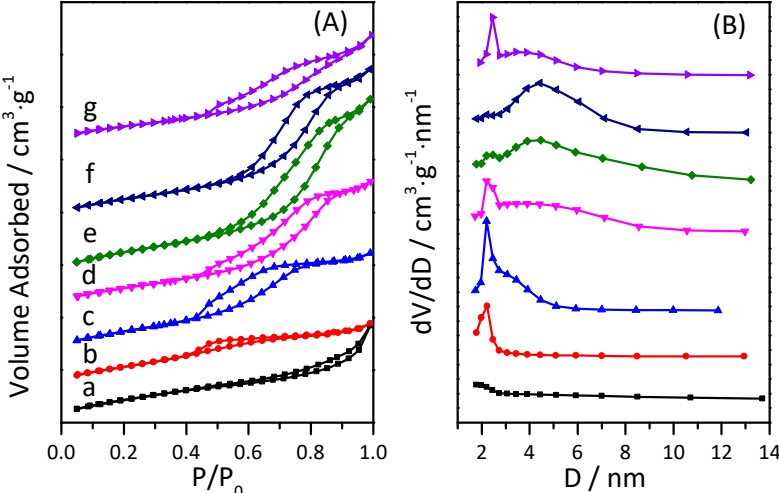

**Figure 1.** (**A**) N$_2$ physisorption isotherm and (**B**) pore size distributions of a. PtWZr, b. PtWZr7Ti3, c. PtWZr5Ti5, d. PtWZr4Ti6, e. PtWZr3Ti7, f. PtWZr2Ti8 and g. PtWTi.

**Table 1.** Physicochemical properties of catalysts.

| Catalyst | $S_{BET}$ (m$^2$/g) | $D_{pore}$ (nm) | $V_{pore}$ (cm$^3$/g) |
|---|---|---|---|
| PtWZr | 57.0 | 2.1 | 0.023 |
| PtWZr7Ti3 | 50.6 | 2.6 | 0.016 |
| PtWZr5Ti5 | 58.6 | 2.8 | 0.040 |
| PtWZr4Ti6 | 59.1 | 3.9 | 0.058 |
| PtWZr3Ti7 | 64.4 | 4.7 | 0.083 |
| PtWZr2Ti8 | 58.9 | 4.4 | 0.071 |
| PtWTi | 47.0 | 3.8 | 0.042 |

The N$_2$ physisorption isotherms and pore size distributions of the Pt/WO$_x$-ZrO$_2$-TiO$_2$ (PtWZrTi) catalysts with different Ti/Zr molar ratio are shown in Figure 1. As can be seen from Figure 1A, the samples show a type IV adsorption isotherm but had a different hysteresis loop. The Pt-WO$_x$/ZrO$_2$ (PtWZr) catalyst exhibited an H4 hysteresis loop, while H2 hysteresis loop was presented in the Pt-WO$_x$/TiO$_2$(PtWTi) and PtWZrTi catalyst. From Figure 1B, the average pore size distribution was around 2-5 nm for all the samples. The physicochemical properties including surface area, pore size and pore volume, are listed in Table 1. The Pt/WOx-ZrO$_2$-TiO$_2$ catalyst with a Ti/Zr molar ratio of 7/3 (PtWZr3Ti7) possessed a high surface area (64.4 m$^2$ g$^{-1}$) and pore volume (0.083 cm$^3$ g$^{-1}$).

Figure 2 shows the XRD patterns of the Pt-WO$_x$/TiO$_2$(PtWTi), Pt/WO$_x$-ZrO$_2$-TiO$_2$ (PtWZrTi) and Pt-WO$_x$/ZrO$_2$ (PtWZr) catalysts. The crystalline phase of the PtWZrTi catalysts was closely related with the different Ti/Zr molar ratio. For the PtWZr catalyst, the Bragg maxima at 30.27, 34.81, 50.38, 60.2 and 62.97° were attributed to ZrO$_2$. The PtWTi catalyst only had the Bragg maxima of anatase TiO$_2$ in Figure 2. With the Ti/Zr molar ratio increasing, the intensity of the ZrO$_2$ maxima decrease while the intensity of TiO$_2$ maxima increased gradually. Interestingly, the new crystalline phase of ZrTiO$_4$ and Ti$_2$ZrO$_6$ was present in the PtWZrTi catalysts, which derived from the reaction between TiO$_2$ and ZrO$_2$ (TiO$_2$ + ZrO$_2$ → ZrTiO$_4$; 2TiO$_2$ + ZrO$_2$ → Ti$_2$ZrO$_6$) [25]. For the PtWZrTi catalyst, with the Ti/Zr molar ratio increasing, it appeared the ZrTiO$_4$ maxima (JCPDS NO. 34-0415) first and then took on the Ti$_2$ZrO$_6$ maxima (JCPDS NO.46-1265). Besides the TiO$_2$, ZrO$_2$, and metallic Pt, the srilankite Ti$_2$ZrO$_6$ was present in the PtWZr3Ti7 catalyst while the PtWZr7Ti3 catalyst had the ZrTiO$_4$. For the all supported Pt catalysts, no obvious diffraction peaks assigned to metallic Pt or WO$_x$ crystal were identified in Figure 2, implying a good dispersion of the Pt particles and W species on the catalyst surface.

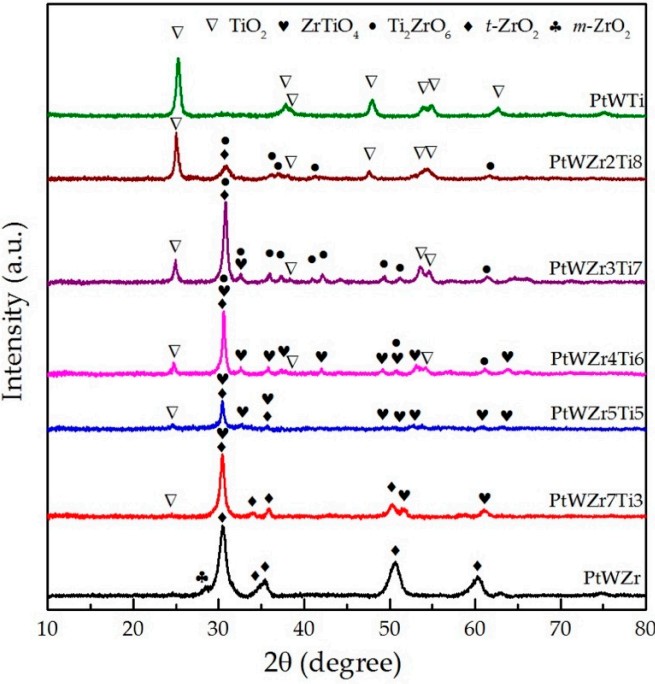

**Figure 2.** XRD patterns the PtWZr, PtWZrTi and PtWTi catalysts.

Figure 3 shows the Raman spectra of the PtWZr, PtWTi and PtWZrTi catalysts. It is found in the PtWZr catalyst that the typical Raman bands at 275, 314, 475 and 647 cm$^{-1}$ due to $t$-ZrO$_2$ [26], while the bands due to $m$-ZrO$_2$ were not observed because of its little content in the sample. The weak and broad band at around 825 cm$^{-1}$ was assigned to the stretching mode of W–O–W bonds, indicating the existence of WO$_x$ nanoparticles stabilized by distorted Zr [26]. The broad band at around 976 cm$^{-1}$ was due to the stretching mode of terminal W=O bonds in WO$_x$, which was always located in the region of 950–1200 cm$^{-1}$ [27]. In the PtWTi catalyst, the peaks of 396, 515 and 636 cm$^{-1}$ were characteristic bands of anatase TiO$_2$. The broad band at 995 cm$^{-1}$ belonging to the terminal W=O stretching mode of WO$_x$ was also observed. For the PtWZrTi catalyst, more or less, the Raman bands of ZrO$_2$, TiO$_2$ and WO$_x$ species took red or blue shifts. The observed band of TiO$_2$ around 630 cm$^{-1}$ increased with the Zr/Ti molar ratio increasing. The band around 800 cm$^{-1}$ with a broad and weak peak was attributed to the A1 vibration mode of Ti–Zr–O (Ti$_2$ZrO$_6$) [28]. Peculiarly, the PtWZr7Ti3 and PtWZr5Ti5 catalyst had the too weak Raman bands to be observed. WOx species were not observed in the XRD but in the Raman spectra. In addition, the Raman results corresponding to TiO$_2$ and ZrO$_2$ consisted of the XRD patterns (Figure 2).

Figure 4 shows the TEM images of the PtWZr, PtWZr3Ti7 and PtWTi catalyst. As can be seen from Figure 4a, Pt particles dispersed on the supports of the PtWTi and PtWZr3Ti7 catalysts. However, the PtWZr catalyst did not have a good dispersion for Pt particles (Figure 4c), which might be caused by the aggregation of Pt particles on the catalysts surface. The Pt particle size centered at 3.5–4 nm for the PtWZr3Ti7 catalyst was smaller than that of the PtWZr catalyst (5.0–5.5 nm). The smaller Pt particle size on the supports was helpful to glycerol hydrogenolysis, it is because the desorption of hydrogen species and sequential spillover to the supports with lower activation energy need the metallic Pt sites with a smaller particle size [29–31].

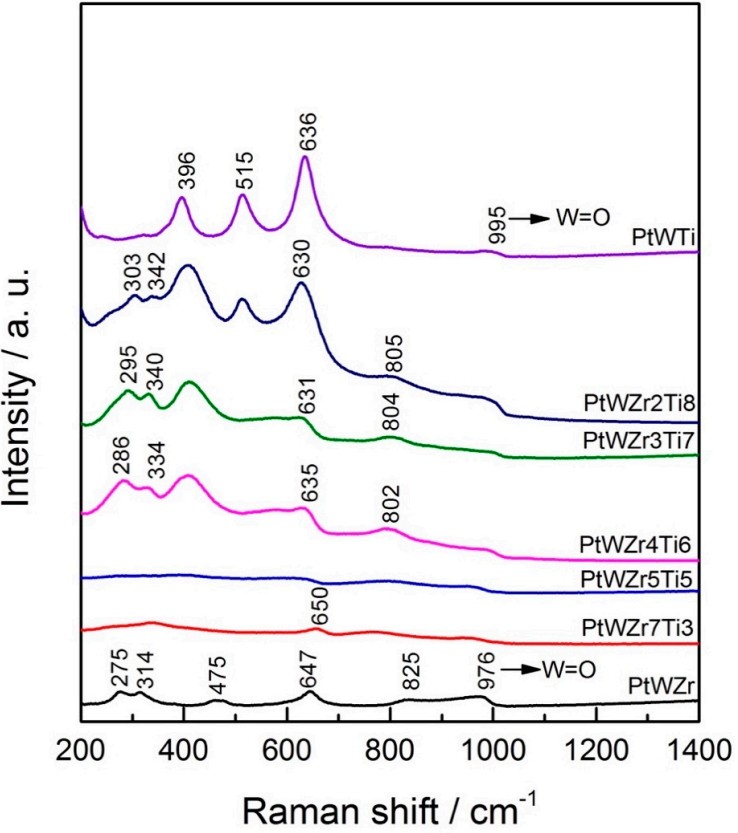

**Figure 3.** Raman spectra of the PtWZr, PtWZrTi and PtWTi catalysts.

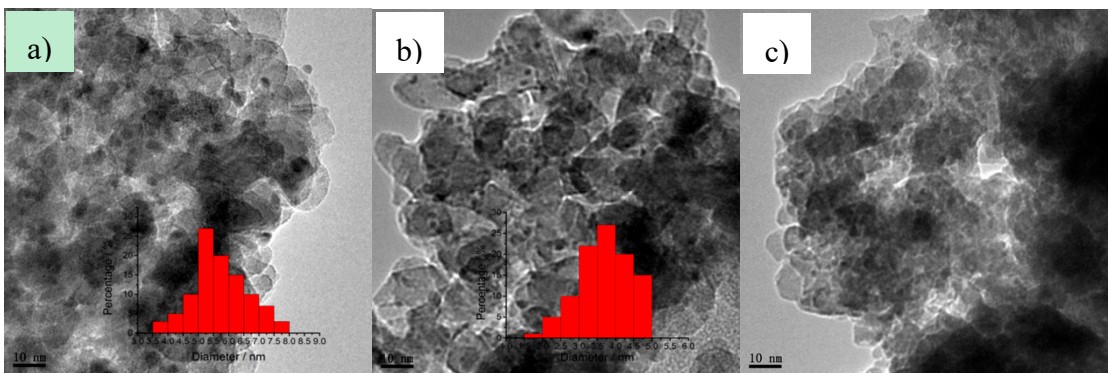

**Figure 4.** TEM images of the (**a**) PtWTi, (**b**) PtWZr3Ti7 and (**c**) PtWZr catalyst.

The surface chemical states and surface element compositions of the PtWZr, PtWZrTi and PtWTi catalysts were characterized by X-ray photoelectron spectroscopy (XPS). Figure 5 shows the XPS spectra in the W4f, Ti3p, Zr4p, Pt4f, W5s and O1s region, respectively. The BE (binding energy) values of corresponding peaks, elemental concentration, and $Pt^0/(Pt^{2+}+Pt^0)$ obtained from the deconvolution are summarized in Table 2. As revealed from the analysis of the spectra in Figure 5a, the $W4f_{5/2}$ and $W4f_{7/2}$ peak of PtWZr catalysts were resolved into two peaks at 37.6 and 35.6 eV, respectively, which were assigned to the $W^{6+}$. $W^{6+}$ species are thought to favor the generation of Brønsted acid sites [32]. Aside from the W4f doublet, there are two Zr 4p BEs of ca. 32.0 and 30.4 eV ascribable to the $Zr^{4+}$ species [14]. For the PtWTi catalyst, the BE at 37.1 eV was due to the Ti3p peak of $TiO_2$. The BE value of Ti 3p ($TiO_2$) rose and fell at around 36.7 eV for the PtWZrTi catalysts, maybe due to the transformation of the partly $TiO_2$ into the new species such as $ZrTiO_4$ or $Ti_2ZrO_6$. Figure 5b presents the XPS spectra of the Pt4f and W5s region, the fitting results are compiled in Table 2. The $Pt^{2+}$ species on the PtWZrTi catalyst with dispersed Pt nanoparticles was observed in Figure 5b and interpreted as an indication of the

strong interaction between Pt and the surface tungsten species or supports [33]. Since the Pt 4f doublet overlapped with the W 5s signal, the latter was subtracted when fitting the Pt 4f spectra (Figure 5b). The Pt 4f spectra of the PtWZr catalyst were fitted into two doublets with the Pt 4f7/2 BEs of ca. 70.8 and 72.4 eV ascribable to $Pt^0$ and $Pt^{2+}$, respectively [34]. Although the identical Pt and W loading on the all sample, the surface Pt and W concentration substantially had some differences with each other, which could result in the different dispersion of active Pt or W species on the catalyst surface.

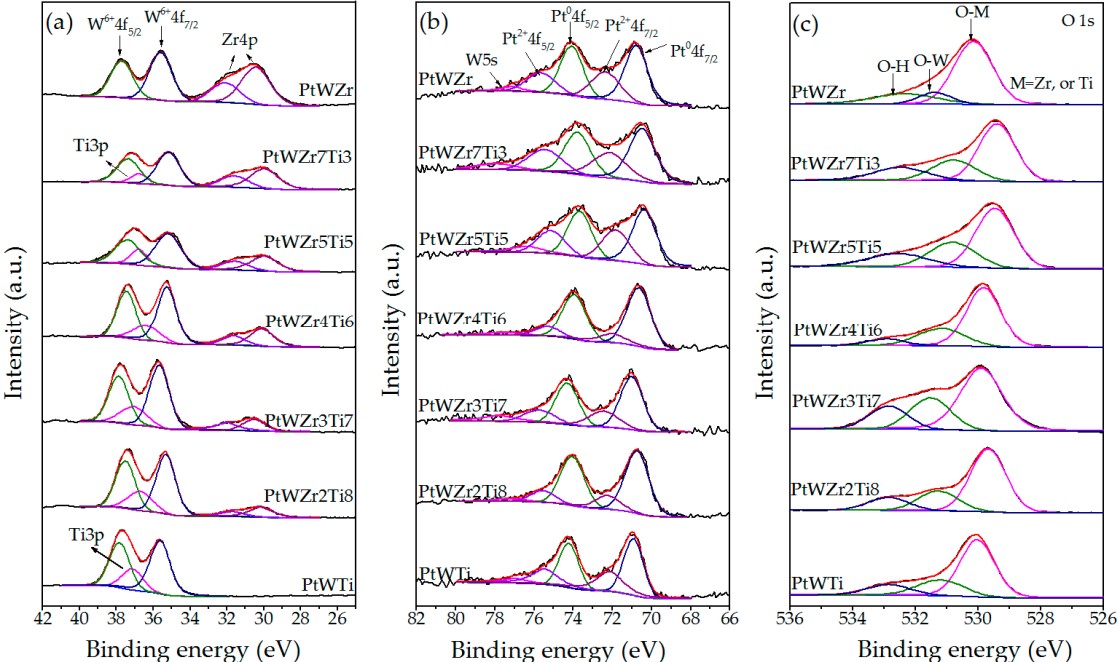

**Figure 5.** XPS spectra of (**a**) W4f, Ti3p, Zr4p, (**b**) Pt4f, W5s and (**c**) O1s for the PtWZr, PtWZrTi and PtWTi catalysts.

**Table 2.** XPS results for PtWZr, PtWZrTi and PtWTi catalysts.

| Catalysts | Binding Energy (eV) | | | | | Surface Elemental Concentration (%) | | $Pt^0/(Pt^{2+}+Pt^0)$ | Zr/Ti |
|---|---|---|---|---|---|---|---|---|---|
| | $W^{6+}4f_{7/2}$ | $Pt^{2+}4f_{7/2}$ | $Pt^04f_{7/2}$ | Zr3d | Ti3p | Pt | W | | |
| PtWZr | 35.6 | 72.4 | 70.8 | 180.1 | - | 0.71 | 1.59 | 0.654 | - |
| PtWZr7Ti3 | 35.2 | 72.1 | 70.5 | 180.1 | 36.7 | 0.58 | 1.32 | 0.612 | 3.42(2.33) |
| PtWZr5Ti5 | 35.2 | 71.8 | 70.4 | 180.0 | 36.8 | 0.76 | 1.72 | 0.630 | 1.43(1) |
| PtWZr4Ti6 | 35.3 | 72.0 | 70.6 | 182.9 | 36.4 | 0.59 | 1.42 | 0.702 | 0.94(0.67) |
| PtWZr3Ti7 | 35.7 | 72.4 | 71.0 | 181.7 | 37.1 | 0.78 | 1.72 | 0.801 | 0.74(0.43) |
| PtWZr2Ti8 | 35.3 | 72.2 | 70.7 | 182.2 | 36.7 | 0.43 | 1.71 | 0.799 | 0.46(0.25) |
| PtWTi | 35.6 | 72.1 | 70.9 | - | 37.1 | 0.37 | 2.16 | 0.610 | - |

Figure 5c shows the XPS spectra of the O 1s region. For the PtWZr catalyst, three BEs at 530.1, 531.4 and 532.8 eV present in the O 1s region, which are attributed to the bonding configuration of O with metal (Zr), W and H, respectively [26]. Comparing with the PtWZr catalyst, these bands of the O1s region in the PtWZrTi catalysts took a shifting, especially referred to the O–M bands. As can be seen from the Figure 5c, interestingly, the intensity of fitting peaks due to O–H in the PtWZr3Ti7 catalyst was the strongest among the all catalysts, reflecting presence of the largest OH groups on the catalyst surface.

In addition, the surface $Pt^0/(Pt^{2+}+Pt^0)$ and Zr/Ti ratios on the PtWZrTi catalyst calculated by the XPS fitting peak results were summarized in Table 2. The PtWZr3Ti7 catalyst had the highest $Pt^0/(Pt^{2+}+Pt^0)$ ratios, indicating the more Pt nanoparticles dispersed on the catalyst surface, which is helpful to the glycerol hydrogenolysis. For all PtWZrTi catalysts, the surface Zr/Ti ratios detected by XPS were substantially higher than the values of theoretical stoichiometry (within bracket of columns

10 in Table 2). That is because a slight aggregation of Zr species on the catalysts surface, the same findings also can be observed by Chaudhary that the preferential complexation between W species and $Zr^{4+}$ as W species will be expelled to the surface [35].

*2.2. Acidic Properties*

Pyridine infrared spectroscopy (Py-IR) is usually used to determine the nature and number of acid sites [36]. The concentration of Brønsted and Lewis acid sites of PtWZr, PtWZrTi and PtWTi catalysts were calculated by the integral intensities of the typical bands centered at 1540 and 1450 $cm^{-1}$, and the amounts of Brønsted and Lewis acid sites at pyridine evacuation temperatures of 200, 300 and 400 °C were summarized in Table 3, respectively. As can be seen from Table 3, all the catalysts show more Lewis acid sites than Brønsted acid sites and the B/L ratios were close to 0.44–0.49. Compared with the PtWZr and/or PtWTi catalysts, the PtWZrTi catalysts had more Lewis and Brønsted acid sites, it is indicated that combination of $ZrO_2$ and $TiO_2$ improved the surface acid sites amount of the Pt-$WO_x$ catalyst. With the Ti/Zr ratio increasing, the Brønsted and Lewis acid sites both increased first and then decreased at the temperatures of 200, 300 and 400 °C. The PtWZrTi catalysts had higher Brønsted acid sites than that of the PtWZr and PtWTi catalysts. With the $ZrO_2$-$TiO_2$ complex as the support, some Zr atoms were replaced by Ti, the electron density of M–O–W (M=Ti, Zr) bonds would be lowered due to the higher electronegativity of Ti than Zr [37] and there would be more Brønsted acid sites than in $ZrO_2$ alone or $TiO_2$ support. In addition, higher surface areas of the PtWZrTi catalysts (Table 1) promoted the dispersion of $WO_x$ species, which can supply more Brønsted acid sites. In particular, when the Zr/Ti ratio reached up to 3/7, the PtWZr3Ti7 catalyst had the most Lewis acid sites among the PtWZrTi catalysts. The reason is not only that it had the highest surface area, but also due to the presence of the large $Ti^{4+}$ species ($Ti_2ZrO_6$) as the Lewis acid sites on the PtWZr3Ti7 catalyst surface.

**Table 3.** Surface Lewis and Brønsted acid sites ($\mu mol/g^{-1}$) of the prepared catalysts.

| Catalysts | B Acid Sites/$\nu_{19b}$ = 1540 $cm^{-1}$ | | | L Acid Sites/$\nu_{19b}$ = 1450 $cm^{-1}$ | | | B/L |
|---|---|---|---|---|---|---|---|
| | 200 °C | 300 °C | 400 °C | 200 °C | 300 °C | 400 °C | |
| PtWZr | 42.6 | 29.4 | 10.3 | 90.1 | 52.3 | 35.1 | 0.46 |
| PtWZr7Ti3 | 48.5 | 32.6 | 12.2 | 97.8 | 56.3 | 37.0 | 0.49 |
| PtWZr5Ti5 | 50.2 | 34.1 | 13.4 | 101.6 | 59.0 | 38.8 | 0.49 |
| PtWZr4Ti6 | 50.8 | 33.9 | 13.1 | 101.5 | 63.3 | 39.2 | 0.48 |
| PtWZr3Ti7 | 51.4 | 33.3 | 13.9 | 101.7 | 69.0 | 40.5 | 0.45 |
| PtWZr2Ti8 | 45.0 | 30.7 | 11.0 | 91.5 | 53.9 | 35.8 | 0.48 |
| PtWTi | 32.0 | 18.9 | 7.6 | 68.2 | 39.5 | 24.8 | 0.44 |

*2.3. Catalytic Performance*

The activity of the PtWZr, PtWZrTi and PtWTi catalysts and products yield for glycerol hydrogenolysis are shown in Figure 6. The PtWTi catalyst exhibited the lowest glycerol conversion of 26.3% and PtWZr catalyst shows the one of 53.4%. It can be observed that the PtWZrTi catalysts possess higher conversion than the PtWZr or PtWTi catalyst. According to the round curves in Figure 6A, the glycerol conversion increased first and then decreased with the Ti/Zr ratio increasing. The product selectivity for the PtWZrTi catalysts in an order as follows: 1-propanol (1-PO) > 1,3-PDO > 2-propanol (2-PO) > 1,2-propanediol (1,2-PDO). For all the catalysts, the 1-PO was the main product rather than 1,3-PDO or 2-PO, which is because the Lewis acid sites on the catalysts surface were more than the Brønsted acid sites for all the catalysts. The Lewis acid sites are thought to play a key role in the selective conversion of glycerol towards 1-PO [38,39]. The PtWZr3Ti7 catalyst had the highest activity, with a conversion of 73.8%, and the highest selectivity to 1-PO of 57.3%. Figure 6B shows the yield of main products 1-PO and 1,3-PDO for the PtWZr, PtWZrTi and PtWTi catalysts. Comparatively, the PtWZr catalyst had a lower yield of 1-PO and 1,3-PDO than that of PtWZrTi catalysts, while it was higher than that of PtWTi catalysts. For the PtWZrTi catalysts, the yield of 1-PO and 1,3-PDO both

increased first and then decreased with the Ti/Zr increasing. The PtWZr3Ti7 catalyst performed the highest catalytic performance, with a 1,3-PDO yield of 25.3% and a 1-PO yield of 42.3%.

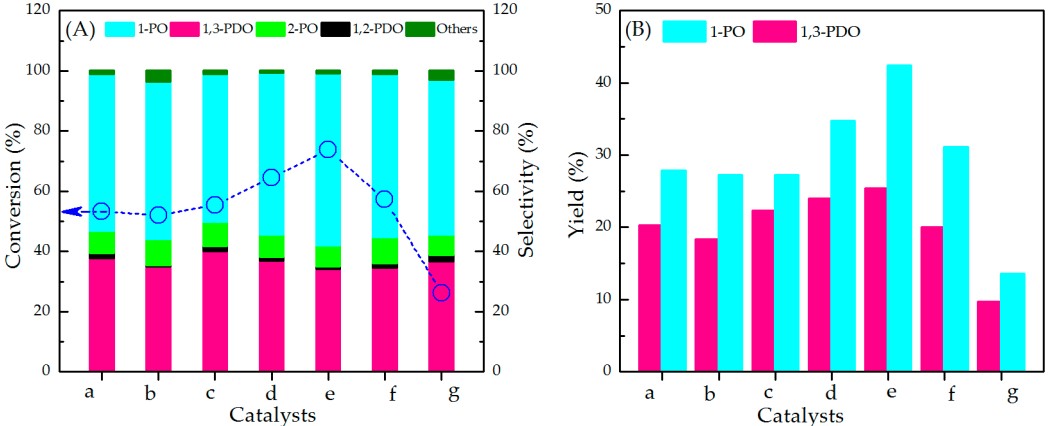

**Figure 6.** (**A**) Glycerol conversion (circle) and selectivity (bar) for the PtWZr, PtWTi and PtWZrTi catalysts and (**B**) yield of 1,3-PDO and 1-PO. a: PtWZr, b: PtWZr7Ti3, c: PtWZr5Ti5, d: PtWZr4Ti6, e: PtWZr3Ti7, f: PtWZr2Ti8and and g: PtWTi. Reaction conditions: 140 °C, 5 MPa, $H_2$:glycerol:$H_2O$ = 10:1:5.1 (molar ratio), flow rate of 50 wt % aqueous solution was 0.03 mL/min, reaction time of 15 h.

In summary, the surface acid sites played an important role on the glycerol hydrogenolysis. The surface acid sites including Lewis and Brønsted acid in Pt-WOx based catalysts were improved by combination of $ZrO_2$ and $TiO_2$. An increase in acid sites was also obtained by Boffito when adding $TiO_2$ into sulfated $ZrO_2$ for the preparation of sulfated $ZrO_2$-$TiO_2$, and this can be attributed to the charge imbalance caused by heteroatom linkage [40]. No doubt, the more acid sites on the catalysts were beneficial to obtain higher conversion of glycerol hydrogenolysis. The Pt-WOx based catalysts supported on $ZrO_2$-$TiO_2$ complex had higher activity of glycerol hydrogenolysis than supported on $ZrO_2$ or $TiO_2$. It is interestingly found that the optimized Ti/Zr molar ratio was 7/3 for the Pt/WOx-$ZrO_2$-$TiO_2$ catalysts, and the PtWZr3Ti7 catalyst performed the highest conversion of glycerol hydrogenolysis, with a 1,3-PDO yield of 25.3% and a 1-PO yield of 42.3%, which was attributed to the highest surface acid sites. In addition, the high concentration of $Pt^0$ and small Pt particles (3.5–4 nm) of catalysts surface were in favor to the glycerol hydrogenolysis.

## 3. Materials and Methods

### 3.1. Catalyst Preparation

The Pt/WOx-$ZrO_2$-$TiO_2$ catalysts were prepared by an Evaporation Induced Self Assembly method using $EO_{20}PO_{70}EO_{20}$ (PluronicP123, Sigma, Shanghai, China) as a template. The obtained catalysts were named as PtWZr$_x$Ti$_y$ where y/x means the Ti/Zr molar ratio and the W content was fixed at 15%. For example, the synthesis of PtWZr3Ti7 was as follows: 6.0 g P123 was firstly dissolved in a solution containing 150 mL ethanol and 10 mL acetic acid with rapidly stirring until a homogeneous transparent solution was formed. Then 8.92 mL zirconium propoxide ($Zr(OC_3H_7)_4$; Aladdin, Shanghai, China) and 15.8 mL titanium butoxide ($C_{16}H_{36}O_4Ti$; Aladdin) were added slowly and stirred for about 30 min. Required amounts of 12-phosphotunstic acid (Sinopharm Chemical Reagent Co., Ltd, Shanghai, China) was first dissolved in 10 mL ethanol and then was added drop by drop into the above solution. After further stirring for 10 min, the obtained solution was poured into Petri dishes (diameter of 90 mm) and transferred into an oven at 60 °C for solvent evaporation for about 24 h. Then the as-synthesized solid was collected, grinded and calcined at 600 °C in flowing air for 5 h with a ramping rate of 1 °C/min. Then the WO$_x$-$ZrO_2$-$TiO_2$ complex was impregnated with an aqueous solution of $H_2PtCl_6·6H_2O$ (Sinopharm Chemical Reagent Co., Ltd, Shanghai, China) for 12 h, dried at

110 °C overnight and then was calcined in static air at 450 °C for 3 h. The final obtained catalysts were designated as PtWZr3Ti7, of which the Pt nominal mass loading in the catalyst was fixed at 3.0%.

### 3.2. Characterization

All the PtWZrxTiy catalysts were pre-reduced in $H_2$ atmosphere before characterization. $N_2$ adsorption and desorption isotherms were measured on a Micrometrics Tristar 3020 at −196 °C. The surface area was calculated using the BET method and the pore size distributions were obtained by the BJH method according to the desorption branch of the isotherms. The samples were degassed under vacuum at 200 °C for 2 h before the physisorption measurements. The XRD (powder X-ray Diffraction) patterns were recorded in 2θ values ranging from 10 to 80° at a 2 °/min scan speed on a D8 Advance X-ray diffractometer (Bruker, Bremen, Germany) with a Cu Kα radiation. Raman spectroscopy was detected on a Renishaw via a microscope with an Ar ion laser employing a 514 nm laser excitation. The spectra of the samples were recorded at ambient condition within the 100–1200 $cm^{-1}$ region. $NH_3$-TPD was carried out on a Tianjin Pengxiang Chemisorption Analyzer equipped with a thermal conductivity detector (TCD). The samples were pretreated in He at 500 °C for 1 h and then cooled to 100 °C followed by saturated adsorption with pure $NH_3$ for about 30 min. The samples were then heated to 600 °C at a rate of 5 °C/min and the desorbed $NH_3$ was detected according to the TCD signal. Pyridine adsorption FT-IR was adopted to determine the type of acid sites. Samples were preheated at 400 °C under vacuum for 2 h, and then cooled to 200 °C. After the pyridine adsorption for 10 min and balance for 5 min, the excess pyridine was outgassed under low vacuum for 10 min and high vacuum for 30 min and the FT-IR signal was detected. The temperature was increased to 300 °C and 400 °C for IR scanning respectively. TEM and HRTEM images were recorded on a transmission electron microscope (TEM, Philips, Tecnai 20) operating with an accelerating voltage of 200 kV. XPS was collected on an AXIS Ultra DLD instrument (Amsterdam, Netherlands). The signals were calibrated by the polluted C1s electron binding energy peak at 284.6 eV.

### 3.3. Catalysts Test

The hydrogenolysis of glycerol reaction was conducted in a vertical stainless steel fixed-bed reactor. Four grams of catalyst was placed at the constant temperature section of the reactor and both ends of which were packed with quartz balls. Prior to the reaction, the catalyst was reduced at 250 °C with a constant $H_2$ flow (50 mL/min) for 2 h. Then the reactor was cooled down to designated reaction temperature (140 °C) and 50 wt % glycerol aqueous solution was pumped continuously into the reactor at 0.03 mL/min rate under 5 MPa $H_2$ pressure and the $H_2$ flow rate was fixed to maintain $H_2$:glycerol = 10:1 (molar ratio).

The liquid products were collected after reaction time of 15 h and were analyzed using a gas chromatography (Agilent 4890D, Santa Clara, CA USA) equipped with a FID detector and a CATALOG 19091N036 capillary column (60 m × 0.250 mm). 1,4-butanediol was used as internal standard.

The conversion of glycerol, selectivity and yield of the products were calculated by the following equations:

$$Conversion\ of\ glycerol = [moles\ of\ glycerol(in)\text{-}moles\ of\ glycerol(out)]/[moles\ of\ glycerol(in)] \times 100\%$$

$$Selectivity\ of\ one\ product = (moles\ of\ one\ product)/(moles\ of\ all\ products) \times 100\%$$

$$Yield = (Conversion\ of\ glycerol) \times (Selectivity\ of\ one\ product) \times 100\%$$

## 4. Conclusions

A series of Pt/WOx-$ZrO_2$-$TiO_2$ catalysts with different Ti/Zr molar ratios were prepared and used for efficient hydrogenolysis of glycerol to 1-PO and 1,3-PDO. The effect of Ti/Zr molar ratios of the support $ZrO_2$-$TiO_2$ on structural and acidic properties of the catalysts was investigated by BET, XRD, Raman, TEM, XPS and Py-IR characterization. The PtWZrTi catalysts had a high surface area,



which could improve the dispersion of WOx species on the supports surface. Two new crystalline phases of $ZrTiO_4$ and $Ti_2ZrO_6$ were observed in the PtWZrTi catalysts, the more $Zr^{4+}$ ($ZrTiO_4$) and $Ti^{4+}$ ($Ti_2ZrO_6$) resulted in the more Lewis acid sites compared with the PtWZr and PtWTi catalysts. The more Lewis acid sites on the catalyst surface resulted in 1-PO as a dominant product of glycerol hydrogenolysis over the supported Pt-WOx catalysts. The $Pt/WOx-ZrO_2-TiO_2$ catalyst with a Ti/Zr ratio of 7/3 (PtWZr3Ti7) exhibited the highest 1,3-PDO yield (25.3%) and 1-PO yield (42.3%), due to its more acid sites including Brønsted and Lewis, and higher concentration of $Pt^0$.

**Author Contributions:** Investigation: W.J. and Z.X.; Methodology: Z.X., Z.H., J.L. and F.H.; Writing—original draft: W.J. and Z.X.; Writing—reviewing and editing: W.J. and Z.Z.; Supervision: Z.Z.; All authors have read and agreed to the published version of the manuscript.

**Funding:** This research was funded by National Natural Science Foundation of China (21802115, 21603069, 91534115), Science and Technology commission of Shanghai Municipality (14DZ2261100). Science and Technology Program of Hunan Province of China (2017XK2048 and 2018JJ3501).

**Acknowledgments:** This work was financially supported by National Natural Science Foundation of China (21802115, 21603069, 91534115), Science and Technology commission of Shanghai Municipality (14DZ2261100). Science and Technology Program of Hunan Province of China (2017XK2048 and 2018JJ3501).

**Conflicts of Interest:** The authors declare no conflict of interest.

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
