# Peer review of "Hydrogenolysis of Glycerol on the ZrO2-TiO2 Supported Pt-WOx Catalyst"

_catalysts, doi:10.3390/catal10030312_

Round 1
Reviewer 1 Report
Comments to Authors
The reviewed paper entitled “Hydrogenolysis of Glycerol on the ZrO2-TiO2 Supported Pt-WOx Catalyst” (manuscript no. catalysts-736629) by Zhiwen Xi, Zhe Hong, Fangtao Huang, Junhui Li, Zhirong Zhu and Wenzhi Jia is mainly devoted to the synthesis of the complex supported system based on platinum and tungsten as active components and to the verification of its catalytic activity in hydrogenolysis of glycerol. The paper is interesting and generally well written. Both the presented topic and the reported results are quite important for a catalytic community. I feel, that after minor revision this paper should be considered for publication in Catalysts. The more specific remarks are listed below.
General remarks:
- English usage has to be checked. A few elementary errors are noticeable, e.g. Abstract, line 12: A series of Pt-WOx/ZrO2-TiO2 catalysts with different Ti/Zr molar ratios were prepared instead of was prepared; Introduction, p. 26: the utilization of biomass as a sustainable alternative rapidly pay people’s attention, instead of pays people’s attention; Introduction, p. 35: Numerous works has been done, instead of have been done; Introduction, p. 39-40: catalysts were mostly investigated due to its potential application prospect, instead of due to their potential application prospect; Section 2.1, line 117: The observed band of TiO2 around 630 cm-1 increase, instead of increases; Section 2.1, lines 128-129: Pt particle size centred at 3.5-4 nm for the PtWZr3Ti7 catalyst are smaller than that, instead of is smaller than that; Section 2.1, lines 164-165: which is help to the glycerol hydrogenolysis, instead of which is helpful or which helps; Section 2.1, line 218: No double, the more acid sites (…), instead of No doubt; Section 2.1, line 286: The PtWZrTi catalysts has, instead of have, etc.
- The link between the catalyst structure and its reactivity is not fully exploited in the text. This system is rather complex, thus not all dependences can be analyzed in detail in the paper focused more strongly on catalytic properties. The investigated system requires however a separate paper describing its solid state properties;
- Relatively high level of generality is unfortunately characteristic of this paper, there are however some sentences or fragments, which are too vague to be true/clear, e.g. Abstract, line 17: (…) the more acid sites were detected in the Pt-WOx/ZrO2-TiO2 catalysts; Section 2.1, lines 84-86: The above results indicate that the different Ti/Zr molar ratio affect the physicochemical properties of the Pt- WOx/ZrO2-TiO2 catalysts; Section 2.1, lines 92-93: The crystalline phase in the PtWZrTi catalysts has an affinity for the different Ti/Zr molar ratio; Section 2.1, lines 111-112: (…) indicating the existence of distorted Zr stabilized WOx nanoparticles; Section 2.1, line 117: The observed band of TiO2 around 630 cm-1 increase with the Zr/Ti molar ratio decreasing.
- The information provided in the title: ZrO2-TiO2 supported Pt-WOx and reflected in the notation: Pt-WOx/ZrO2-TiO2, seems to be confusing if confronted with Authors’ statement in Experimental, saying that zirconium propoxide and titanium butoxide were added slowly and stirred for about 30 min and the required amounts of 12-phosphotunstic acid was first dissolved in 10 mL ethanol and then was added drop by drop into the above solution. This clearly means that tungsten species can be incorporated into ZrO2-TiO2 binary system structure and cannot be considered as surface active components only.
- Some references are obviously missing, e.g. Introduction, lines: 32-34; Section 2.1, line 99 (after bracket); Section 2.1, lines: 169-179;
- Precisely formulated aim of work is missing.
Minor remarks:
- XRD is not a spectroscopy, thus we tend to use a term “maxima” or “Bragg maxima” and not “peaks”;
- tetragonal and cubic polymorphs of zirconia cannot be distinguished by XRD, thus it is not reasonable to claim that XRD revealed t-ZrO2; fortunately RS provides much more specific attribution;
Author Response
General remarks:
1. English usage has to be checked. A few elementary errors are noticeable, e.g. Abstract, line 12: A series of Pt-WOx/ZrO2-TiO2catalysts with different Ti/Zr molar ratios were prepared instead of was prepared; Introduction, p. 26: the utilization of biomass as a sustainable alternative rapidly pay people’s attention, instead of pays people’s attention; Introduction, p. 35: Numerous works has been done, instead of have been done; Introduction, p. 39-40: catalysts were mostly investigated due to its potential application prospect, instead of due to their potential application prospect; Section 2.1, line 117: The observed band of TiO2 around 630 cm-1 increase, instead of increases; Section 2.1, lines 128-129: Pt particle size centred at 3.5-4 nm for the PtWZr3Ti7 catalyst are smaller than that, instead of is smaller than that; Section 2.1, lines 164-165: which is help to the glycerol hydrogenolysis, instead of which is helpful or which helps; Section 2.1, line 218: No double, the more acid sites (…), instead of No doubt; Section 2.1, line 286: The PtWZrTi catalysts has, instead of have, etc.
Response: Thanks for the reviewer’s comment. English writing is improved according to the reviewer’s advice, the modifications are marked by red fonts in the revised manuscripts.
2. The link between the catalyst structure and its reactivity is not fully exploited in the text. This system is rather complex, thus not all dependences can be analyzed in detail in the paper focused more strongly on catalytic properties. The investigated system requires however a separate paper describing its solid state properties;
Response: It is well known that the relationship between catalyst structure and activity is complex subject in the heterogenous fields. The reported Pt/WOx-TiO2 and Pt/WOx-ZrO2 catalysts have a good reactivity in the references, so we have designed the combination of TiO2 and ZrO2 complex to further improve the reactivity. Effect of the different Ti/Zr molar ratios on the reactivity of Pt/WOx-TiO2-ZrO2 catalysts is investigated, it is found the catalysts with the Ti/Zr molar ratio of 7/3 has an excellent catalytic performance for glycerol hydrogenolysis. Thus, it is necessary to characterize the Pt/WOx-TiO2-ZrO2 catalysts, and their structures related with the reactivity need to be analyzed and investigated. The surface acid sits play an important role on the reactivity, and the surface area can be a factor which affects the glycerol hydrogenolysis.
3. Relatively high level of generality is unfortunately characteristic of this paper, there are however some sentences or fragments, which are too vague to be true/clear, e.g. Abstract, line 17: (…) the more acid sites were detected in the Pt-WOx/ZrO2-TiO2 catalysts; Section 2.1, lines 84-86: The above results indicate that the different Ti/Zr molar ratio affect the physicochemical properties of the Pt- WOx/ZrO2-TiO2 catalysts; Section 2.1, lines 92-93: The crystalline phase in the PtWZrTi catalysts has an affinity for the different Ti/Zr molar ratio; Section 2.1, lines 111-112: (…) indicating the existence of distorted Zr stabilized WOx nanoparticles; Section 2.1, line 117: The observed band of TiO2 around 630 cm-1 increase with the Zr/Ti molar ratio decreasing.
Response: There are some describing analysis maybe produce an ambiguity for the readers. The relative modification or rewriting labeled by green font in the revised manuscript, in terms of the reviewer’s comments point by point.
4. The information provided in the title: ZrO2-TiO2supported Pt-WOxand reflected in the notation: Pt-WOx/ZrO2-TiO2, seems to be confusing if confronted with Authors’ statement in Experimental, saying that zirconium propoxide and titanium butoxide were added slowly and stirred for about 30 min and the required amounts of 12-phosphotunstic acid was first dissolved in 10 mL ethanol and then was added drop by drop into the above solution. This clearly means that tungsten species can be incorporated into ZrO2-TiO2 binary system structure and cannot be considered as surface active components only.
Response: Although the WOx specie incorporated into ZrO2-TiO2 binary system structure, it supplies the indispensable acid sites for catalytic glycerol hydrogenolysis to propyl alcohol, especially Bronsted acid sites. Most of literatures reported the WOx and Pt as the bifunctional active sites of glycerol hydrogenolysis. To avoid the misunderstand, we have corrected the catalysts expression, using Pt/WOx-ZrO2-TiO2 instead of Pt-WOx/ZrO2-TiO2 in the revised text.
5. Some references are obviously missing, e.g. Introduction, lines: 32-34; Section 2.1, line 99 (after bracket); Section 2.1, lines: 169-179;
Response: Thanks for the reviewer’s comment. The references are added in the revised manuscripts.
6. Precisely formulated aim of work is missing.
Response: 1-PO and 1,3-PDO are the important chemical raw materials. Hydrogenolysis of glycerol to produce two important products is significant, the development of high efficient catalyst favored to obtain lots of 1-PO and 1,3-PDO products is our works key.
Minor remarks:
1. XRD is not a spectroscopy, thus we tend to use a term “maxima” or “Bragg maxima” and not “peaks”;
Response: Thanks for the reviewer’s advice, the term modifications are marked by red fonts in the revised manuscripts.
2. tetragonal and cubic polymorphs of zirconia cannot be distinguished by XRD, thus it is not reasonable to claim that XRD revealed t-ZrO2; fortunately RS provides much more specific attribution;
Response: Thanks for the reviewer’s advice. To reduce the misunderstand, we have deleted the description about t-ZrO2 and m-ZrO2 for XRD characterization in the revised manuscript.

Reviewer 2 Report
Title: Hydrogenolysis of Glycerol on the ZrO2-TiO2 Supported Pt-WOx Catalyst
This manuscript describes catalytic activities of Pt-Wox/ZrO2-TiO2 catalyst in the hydrogenolylsis of glycerol to 1.3-PDA and 1-PDO.
The variation of Ti/Zr ratio is reasonable to control the acidity and finally optimize a molar composition of Ti/Zr. The characterization is well done by using N2-sorption, XRD, Raman, TEM, XPS and Py-IR and the results of characterization was well correlated with catalytic activity used.
The explanation of Figure and Table caption is not sufficient to understand the results of manuscript. I think some explanation should be added more.
Finally, I think that this manuscript can be published in this journal based on the overall paper direction after minor revision.
Author Response
The explanation of Figure and Table caption is not sufficient to understand the results of manuscript. I think some explanation should be added more.
Response: Thanks for the reviewer's comments. We have revised the Figure and Table caption by marking red font in the text.
